# Automatic Recognition of Rice Leaf Diseases Using Transfer Learning

**Chinna Gopi Simhadri** and **Hari Kishan Kondaveeti** *

School of Computer Science & Engineering, VIT-AP University, Beside AP Secretariat,
Vijayawada 522237, Andhra Pradesh, India
* Correspondence: kishan.kondaveeti@gmail.com

**Abstract:** Rice, the world's most extensively cultivated cereal crop, serves as a staple food and energy source for over half of the global population. A variety of abiotic and biotic factors such as weather conditions, soil quality, temperature, insects, pathogens, and viruses can greatly impact the quantity and quality of rice grains. Studies have established that plant infections have a significant impact on rice crops, resulting in substantial financial losses in agriculture. To accurately diagnose and manage the diseases affecting rice plants, plant pathologists are seeking efficient and reliable methods. Traditional disease detection techniques, employed by farmers, involve time-consuming visual inspections and result in inadequate farming practices. With advancements in agricultural technology, the identification of pathogenic organisms in rice plants has become significantly more manageable through techniques such as machine learning and deep learning, which are receiving significant attention in crop disease research. In this paper, we used the transfer learning approach on 15 pre-trained CNN models for the automatic identification of Rice leave diseases. Results showed that the InceptionV3 model is outperforming with an average accuracy of 99.64% with Precision, Recall, F1-Score, and Specificity as 98.23, 98.21, 98.20, and 99.80, and the AlexNet model resulted in poor performance with average accuracy of 97.35% among others.

**Keywords:** transfer learning; rice leaf diseases; plant disease detection; image classification; deep learning; machine learning; convolutional neural network

## 1. Introduction

Rice is one of the most important food crops in the world, providing a staple food and energy source for over half of the global population. This means that it plays a crucial role in feeding a large portion of the world's population and is essential for their survival [1]. As the world population continues to grow, so does the demand for food, making it increasingly important to ensure a stable and bountiful rice harvest. Rice is a crop that is grown and consumed in many countries around the world, and it is an important source of food for a large proportion of the global population, particularly in Asia where it is a dietary staple. Rice is a major source of carbohydrates, proteins, vitamins, and minerals and is the primary source of calories for more than half of the population of world, which lives in Asia.

As a result, ensuring a stable and bountiful rice harvest is crucial for food security. This means that the rice crop must be protected from diseases, pests, and other factors that can cause crop failure. This includes using appropriate farming practices, developing effective disease management strategies, and implementing new technologies to improve crop productivity [2]. Therefore, It is crucial for ensuring food security by maintaining the production of rice, as the world population continues to grow, and the demand for food increases. One way to address this challenge is through precision agriculture, which uses advanced technologies to increase crop yields.

One of the most advanced technologies in precision agriculture is the automated leaf disease diagnosis system. This system is used to identify plant illnesses by analyzing images of infectious leaf diseases. It uses a combination of computer vision, image processing,

machine learning (ML), and deep learning (DL) algorithms to identify diseases. Traditionally, human vision-based models were used to identify leaf diseases, but these methods can be time-consuming and expensive. Additionally, these models rely on the opinions of individuals or specialists to determine their performance. In contrast, the automated leaf disease diagnosis system streamlines the diagnostic process, allowing farmers to make prompt and accurate decisions about the condition of their plants. This can help farmers to more efficiently use resources and improve crop yields [3].

The use of ML and DL models for disease diagnosis in rice plants is an area that has not been extensively studied. Despite the potential benefits, this field has not received much attention. Further research in this area could help improve rice crop yield and mitigate losses due to disease outbreaks. However, the use of these advanced techniques has the potential to significantly improve the efficiency and accuracy of disease detection in rice plants. A convolutional neural networks (CNN) is a deep learning model ideal for visual data processing, consisting of 3 layers: input, hidden, and output [4]. The hidden layer has three sublayers: Convolution, Pooling, and Fully Connected. The CNN has adjustable parameters referred to as weights, enabling it to recognize spatial input relationships and perform classification.In the context of rice plant disease diagnosis, CNNs can be a powerful tool for automating the diagnostic process and increasing the accuracy of disease detection [5]. There are several types of CNN models that are suitable for plant disease detection. Some of the most commonly used models include LeNet, AlexNet, VGGNet, and ResNet [6].

All of these CNN models have been used in various studies for the detection of plant diseases, with the choice of which model to use often depending on the specific task and the available resources [7]. One of the major advantages of using CNNs for rice plant disease diagnosis is the ability to identify diseases at an early stage. Early detection can be important for preventing the spread of disease and minimizing damage to crops. Furthermore, automated diagnosis can help farmers to make more informed decisions about crop management and treatment, which can lead to better crop yields and improved economic outcomes. Overall, The use of CNNs can provide an efficient and accurate diagnosis of rice plant diseases that can help farmers to optimize their crop management strategies and improve the overall quality and yield of their rice crops.

Deep learning models, while powerful, do have some limitations [8]. These include the need for large amounts of labeled data to train, high computational resources, a tendency to overfit the training data, and difficulty in understanding how the model is making predictions. These limitations can be a challenge to overcome, particularly in domains with limited data or resources, and can affect the model's ability to generalize to new data and diagnose and fix errors.

Transfer learning can help to mitigate some of these limitations. Transfer learning is a technique where a pre-trained model, trained on a large dataset of one task, is fine-tuned on a smaller dataset of another task [9]. The pre-trained model can be used as a starting point, allowing the model to learn faster and achieve better performance than training a model from scratch, especially when there is limited data available. Additionally, by fine-tuning a pre-trained model, the model is less likely to overfit the training data, which can lead to improved generalization performance. Also, using pre-trained models can save a lot of computational resources, as the model has already been trained on a large dataset, so it takes less time and computational power to fine-tune it on a new dataset.

Transfer learning enables the utilization of a pre-trained model that has already been trained on a large dataset, to be adapted for use with a new dataset. This can help to reduce the amount of data and computational resources required to train a deep learning model, while also reducing the risk of overfitting, which leads to better generalization performance. The method makes it possible to take advantage of already existing large datasets, so the model can be trained faster and more accurately.

Some of the common rice plant diseases and symptoms were listed in Table 1.

**Table 1.** List of Rice leaf diseases, their symptoms, and their description.

| Name of Disease | Description | Cause | Affected Part of the Plant | Damage |
|---|---|---|---|---|
| Bacterial Leaf Blight (BLB) | Water-soaked lesions on leaves that turn brown and dry, leading to wilting and death of the plant. | Xanthomonas oryzae bacteria | Leaves | Severe yield loss, particularly in humid environments. |
| Brown Spot | Irregularly-shaped brown spots with yellow borders on leaves, reduced grain size and quality. | Cochliobolus miyabeanus fungus | Leaves, panicles, grains | Significant yield loss |
| Hispa | Leaf holes on leaves | Dicladispa armigera insect | Leaves, leaf sheaths | serious yield loss |
| Leaf Blast | Circular or elongated necrotic lesions on leaves, panicles, and glumes, reduced grain size and quality. | Magnaporthe oryzae fungus | Leaves, panicles, grains | Severe yield loss |
| Leaf Scald | Yellowish to brownish lesions on leaf blades, leaf sheath, and leaf collar. | Xanthomonas oryzae pv. oryzicola bacterium | Leaves | Serious yield loss, particularly in warm and humid conditions. |
| Leaf Streak | Circular or elongated tan to brown leaf spots, reduced yield. | Cercospora oryzae fungus | | Moderate yield loss |
| Narrow Brown Spot | Small, rectangular brown spots on leaves, reduced yield. | Raphanus sativus var. nasturtii fungus | Leaves | Moderate yield loss |
| Sheath Blight | Dark-brown to black lesions on leaf sheaths and stems, reduced grain size and quality. | Rhizoctonia solani fungus | Leaf sheath, collar, straws | Moderate yield loss |
| Tungro | Chlorotic and necrotic leaf spots, stunted growth, reduced grain size and quality. | Rice tungro spherical virus (RTSV) and Rice tungro bacilliform virus (RTBV) | Leaves | Severe yield loss |

### 1.1. Need for Automatic Rice Leaf Disease Detection

- Rice is a major food and energy source for over half of the global population, making it a critical crop for food security. Ensuring a stable and bountiful rice harvest is crucial for feeding the world's growing population.
- The growth in population and the increasing demand for food is vital to optimizing crop yield. Rice is one of the most important food crops that feed most of the world's population. thus early detection of diseases will help to increase crop productivity [10].
- Rice is susceptible to a wide range of diseases caused by various pathogens and viruses, which can greatly impact the quantity and quality of rice grains. This can lead to significant financial losses for farmers and a reduction in the global food supply.
- Traditional disease detection techniques, such as visual inspections, are time-consuming and can result in inadequate farming practices. By developing automatic detection models, pathologists can more efficiently and effectively identify and manage rice plant diseases [11].
- Advancements in agricultural technology, such as machine learning and deep learning, have made it possible to create accurate and efficient disease detection models for rice plants. Utilizing these techniques can significantly improve disease management and crop productivity.

### 1.2. Challenges Associated with Automatic Rice Leaf Disease Detection

There are several challenges associated with rice leaf disease detection [12], including:

- Diversity of rice leaf diseases: Rice plants can be affected by a wide range of diseases caused by various pathogens and viruses, which can present with different symptoms and affect different parts of the plant. This can make it challenging to accurately detect and diagnose different diseases.
- Lack of standardized methodologies: There are currently no universally accepted and standardized methodologies for rice leaf disease detection. This can make it difficult to compare the performance of different detection methods and to effectively diagnose diseases in different regions.
- Limited accessibility to technology: many farmers in remote rural areas may have limited access to the technology and resources needed for accurate disease detection. This can make it difficult for them to effectively manage diseases and protect their crops.
- Difficulties in data collection: Collecting large and diverse datasets for training and testing automatic detection models can be challenging, particularly when the images of leaf diseases are not taken in similar conditions and lighting.
- Balancing accuracy and computational complexity: Developing a disease detection model that is both highly accurate and computationally efficient can be challenging. Many existing techniques may be too complex or computationally intensive for practical use.

Research has been undertaken to develop technical and AI-based methods for diagnosing paddy leaf diseases. Table 2 presents various studies in the classification of rice diseases that are discussed in the related literature. In one study, Malathi et al. [13], used deep convolutional neural networks to categorize ten distinct species of rice crop pests. The team utilized transfer learning by updating the ResNet-50 model's hyperparameters and layers, resulting in an improved model with an accuracy rate of 95.012%. Author Sethy P. K et al. [14] evaluated 13 CNN models for rice disease identification using transfer learning and deep features with SVM. The effectiveness of all classification models based on CNN and conventional techniques was compared. The performance of MobileNetv2 + SVM and ResNet50 + SVM are close enough to be comparable. Haridasan et al. [15] proposed a system that uses computer vision and machine learning to automatically detect and classify diseases in rice plants, specifically focusing on 5 common diseases in Indian rice fields. It uses image processing, segmentation, and a combination of an SVM classifier and CNNs to accurately recognize and classify the disease, getting a validation accuracy of 0.9145.

**Table 2.** Related Literature Survey to Classify and identify rice leaf diseases.

| Study & Year | Objective | No of Images | Algorithm/Method | Leaf Diseases | Performance |
|---|---|---|---|---|---|
| Yang et al. [16] & 2023 | Introduced a new model rE-GoogLeNet, which is able to accurately identify rice leaf diseases in natural environments. | 1122 | rE-GoogLeNet | Aphelenchoides besseyi, Leaf blight, Red blight, Leaf smut, Rice sheath blight, Bacterial leaf streak, Brown spot and Rice blast | Avg Accuracy 99.58% |
| Latif et al. [17] & 2022 | Proposed method for identifying and categorizing rice leaf diseases utilizing transfer learning through DCNN. | 2167 | Modified VGG19 | Healthy, Narrow Brown Spot, Leaf Scald, Leaf Blast, Brown Spot, BLB | Avg Accuracy 96.08% Precision = 96.20% F1-score = 96.16% |
| Daniya et al. [18] & 2022 | Introduced an effective optimization deep learning framework ExpRHGSO algorithm for disease detection and classification | 1006 | ExpRHGSO Algorithm | Bacterial Leaf Blight, Blast, and Brown spot | Accuracy = 91.6%, Sensitivity = 92.3%, Specificity = 91.9% |
| Bari et al. [19] & 2021 | Faster R-CNN algorithm proposed RPN architecture | 2400 | Faster R-CNN | Rice blast, Brown spot, and Hispa | Rice blast = 98.09%, Brown Spot = 98.85%, Hispa = 99.17% |
| Islam et al. [20] & 2021 | Proposed an automated detection approach with the deep learning CNNmodel | 984 | VGG-19, InceptionResnetV2, ResNet-101, Xception | Brown Spot, Leaf Blight, Leaf Smut, Bacterial Leaf Blast | Accuracy = 92.68% (Inception-ResNet-V2) |

**Table 2.** *Cont.*

| Study & Year | Objective | No of Images | Algorithm/Method | Leaf Diseases | Performance |
|---|---|---|---|---|---|
| Wang et al. [21] & 2021 | Proposed the ADSNN-BO model based on MobileNet structure and augmented attention mechanism. | 2370 | ADSNN-OB model | Brown spot, hispa, and leaf blast. | Accuracy = 94.64 |
| Rahman et al. [22] & 2020 | Three different training methods compared on state-of-the-art CNN architectures | 1426 | VGG16, InceptionV3, MobileNetv2, NasNetMobile, SqueezeNet, SimpleCNN | False Smut, BPH, BLB, Neck Blast, Stemborer, Hispa, Sheath Blight, Brown Spot | VGG16 = 97.12%, InceptionV3 = 96.37%, MobileNetv2 = 96.12%, NasNetMobile = 96.95%, SqueezeNet = 92.5%, Simple CNN = 94.33% |

Author Chen et al. [23] used deep learning techniques to improve image processing and classification. They combined DenseNet and Inception modules and achieved high accuracy on a public dataset, with an average of 94.07% or higher. Their model also achieved an average accuracy of 98.63% for classifying rice disease images. Lei Feng et al. [24] employed hyperspectral imaging(HSI) to detect paddy leaf diseases and developed a CNN architecture as a classification model using deep transfer learning techniques. They found that fine-tuning was the most efficient solution, achieving 88% accuracy.

In their study, Thenmozhi et al. [25] proposed a deep CNN model that was able to classify different types of insects with high accuracy. They compared their model's performance to other pre-trained transfer learning models and found that their deep CNN model achieved the highest level of accuracy on all three publicly available datasets. Similarly, Upadhyay et al. [26] presented a technique for identifying and classifying rice plant diseases by analyzing lesion characteristics such as shape, color, and size on leaf images. They used a fully connected CNN method and achieved an accuracy rate of 99.7% on the dataset used, demonstrating the effectiveness of this approach. In another study, Hossain et al. [27] proposed a new CNN-based model to recognize rice leaf diseases by reducing network parameters. The model was trained and tested on a dataset consisting of 4199 images of five common rice leaf diseases, achieving high accuracy rates in both training and validation phases. The model was also tested on independent images, with good recognition rates for specific diseases.

Chen et al. [28] introduced a new technique BLSNet, which aims to accurately detect and identify the leaf damage caused by the Rice Bacterial Leaf Streak (BLS) disease, which can greatly affect the quality and quantity of rice growth. BLSNet was compared to other benchmark models and was found to have better performance in identifying the damage. Additionally, it has the potential to be an effective tool for estimating the severity of BLS disease. Stephen et al. [29] conducted a study on identifying healthy and diseased leaves using four CNN architectures. They utilized ResNet34 and ResNet50 to avoid gradient problems and applied self-attention with ResNet18 and ResNet34 to improve feature selection. Their suggested ResNet34 architecture with self-attention attained an accuracy rate of 98.54% in multiclass classification and outperformed other methods. In contrast Li et al. [30] developed a deep learning-based framework for detecting plant leaf diseases and pests in videos. The framework used Faster-RCNN and image-training models to detect low-quality images. Their experiments showed that their custom backbone was better than other frameworks in detecting non-trained videos in the system.

Latif et al. [17] proposed a method for identifying and categorizing rice leaf diseases using transfer learning via DCNN and achieved a 96.08% accuracy. Bari et al. [19] developed a faster R-CNN approach for diagnosing rice leaf diseases. Accuracy was improved by combining a database of healthy/infected leaves with a public database and image augmentations. Results are promising for identifying healthy/infected leaves in the lab and field images. Islam et al. [20] compared four deep learning networks and found that Inception-ResNet-V2 had the highest accuracy (92.68%). The dataset consisted of five image classes, including disease and healthy classes. Inception-ResNet-V2's unique structure showed better adaptability to the data. Transfer learning was used to improve accuracy and simplify training time. The ResNet-101 network had the second-highest accuracy (91.52%).

Overall, the above studies have utilized CNN, DCNN, Faster-RCNN models to classify images of rice leaves displaying various diseases, such as bacterial leaf blight, brown spot, leaf blast and sheath blight. The primary issue is the scarcity of images, where only a limited number of classes have been tested and each class has fewer than 100 images. This shortage of data results in an insufficient representation of the classes and weakens the ability of the model to accurately classify images. This lack of image data presents a significant challenge in image classification as the model needs a sufficient amount of information to learn and make accurate predictions. A limited dataset leads to limited learning and generalization capabilities, resulting in a weakened model that is unable to accurately classify images.

Our proposed approach takes into account these limitations and utilizes a robust dataset with a substantial number of images per class to achieve exceptional accuracy in the classification of diseases affecting rice leaves. This abundance of data provides a comprehensive representation of the classes and strengthens the model's ability to classify images accurately. Furthermore, our implementation of the InceptionV3 model has achieved remarkable accuracy with a score of 99.64%. The use of advanced models such as InceptionV3, which was trained on a large dataset, enables the model to learn complex features and representations, leading to improved performance. This exceptional performance far surpasses previous works and demonstrates the superiority of our proposed method.

### 1.3. Objectives of This Work

The objectives of the research work described in the paper are:

- To apply transfer learning to pre-trained CNN models for identifying rice leaf diseases automatically.
- To enhance the performance of these transfer learning models.
- To assess and compare the performance of different transfer learning models for identifying rice leaf diseases.
- To identify the best performing and most effective transfer learning models for identifying rice leaf diseases.

### 1.4. Primary Contributions of This Work

The primary contributions of the work in this paper are as follows:

- We created and utilized a dataset of 10,080 images of ten rice leaf diseases for our experiment.
- While many researchers focus on a few common rice diseases, we aimed to create a deep transfer learning model that can classify up to 10 diseases namely Bacterial Leaf Blight, Brown Spot, Hispa, Leaf Blast, Leaf Scald, Leaf Streak, Narrow Brown Spot, Sheath Blight, and Tungro. Additionally, we also included a healthy class as part of the classification.
- We employed transfer learning on pre-trained models to enhance their performance in detecting rice leaf diseases.
- The best and most effective model among the 15 pre-trained deep CNN models was identified.

This paper was structured into several sections: Section 2 explains the details of the dataset, pre-trained NN, evaluation methods, and experimental design. Section 3 analyzes the experiment outcomes to evaluate the pre-trained models' efficiency and interpret their impact. Lastly, Section 4 summarizes the paper's main achievements and provides directions for future research.

### 2. Materials and Methods

Feature extraction involves selecting relevant features from raw data to represent it in a meaningful way, while implicit processing involves learning directly from the raw data without explicitly extracting features. Implicit processing using deep learning models can often outperform feature-based methods because the models are able to learn more complex and abstract features that are difficult to extract using traditional feature extraction techniques, so the use of deep learning models eliminates the need for explicit feature extraction or segmentation, as these tasks are implicitly handled by the model. This can greatly simplify the process of identifying and classifying pests and diseases in rice crops. Generally, pre-trained CNN models can be reused for a particular application, but a number of changes must be made to adapt the model to the specific task at hand.

The first step is to update the classification layer, it is typically updated last in a deep learning model, after the earlier layers have learned to extract useful features from the input data. However, during fine-tuning, the classification layer may be updated first to quickly

adapt the model to a new task or dataset, while the earlier layers can still be useful for feature extraction. Once the classification layer has been updated, the earlier layers can be fine-tuned to better extract task-specific features, leading to improved performance on the new task. This will ensure that the model can accurately predict the correct class for each input.

Next, a new layer was added in place of the learnable layer that combines features from earlier layers. The process of replacing a learnable layer involves identifying the layer type and position to replace, instantiating the new layer, inserting it into the network, possibly freezing its parameters, training the network, and evaluating its performance. Depending on the CNN model being used, this could be a convolution2D layer or a fully connected layer. This new layer will help to extract more relevant features from the input image, which can improve the model's performance. For example, if the task is to identify rice leaf diseases, the new layer should be able to extract features such as the shape, size, and color of the lesions.

To improve the efficiency of training, certain layers of the model can be kept unchanged. The best number of layers to be frozen can be decided by experimenting and evaluating the performance of the task and dataset. This approach is taken to prevent the model from becoming too specialized for the training data and to improve its ability to work with new data. In this particular project, all layers were trained thoroughly because the technology made it possible.

In order to get the dataset ready for the CNN model, the images must be resized. This is typically done to ensure that the images have the same dimensions and can be fed into the model. Furthermore, the dataset should be partitioned into training and validation sets. This will allow for efficient training and testing of the model, which is crucial to ensure that the model is able to generalize well to new unseen data.

Finally, the pre-trained CNNs were re-trained using the rice leaf disease dataset, and its performance was measured. This will allow for an evaluation of the model's accuracy and provide insight into the model's strengths and weaknesses. Details on each stage were provided in the following subsections for a more in-depth understanding of the methodology. Overall, the proposed methodology is a powerful tool for identifying and classifying pests and diseases in rice crops, which can help to enhance the yield and quality of the crop. Figure 1 represents a flowchart of each stage of the proposed methodology using transfer learning on pre-trained convolutional neural networks (CNNs).

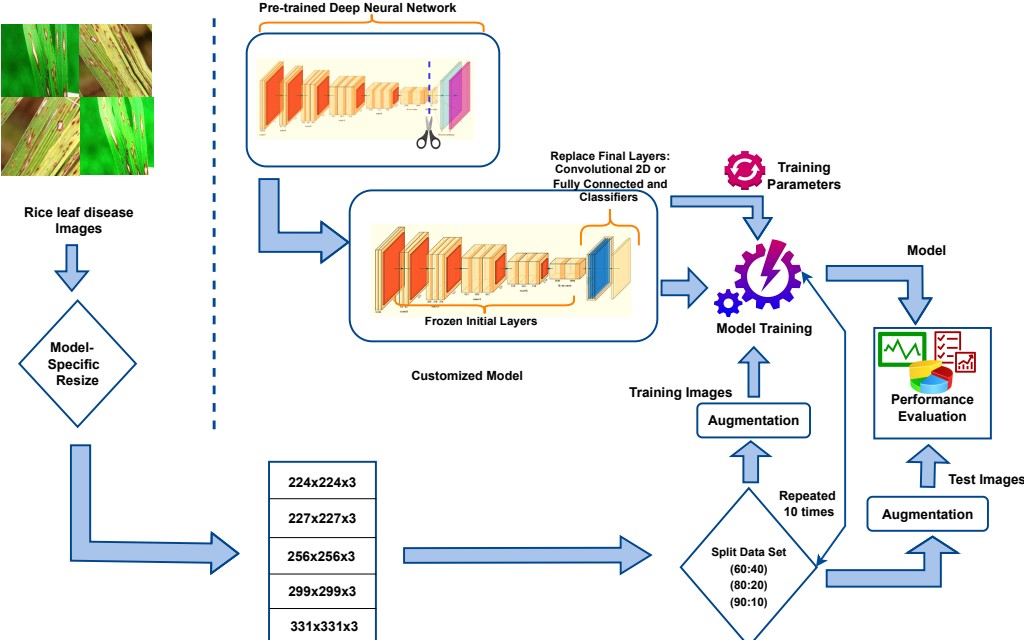

**Figure 1.** An illustration of the successive stages in the conceptualization of the proposed architecture.

*2.1. Dataset*

In this study, we collected publicly available rice leaf disease images of nine diseased classes and one healthy class from Kaggle [31] and Mendeley data [32] and combined them to create a dataset. Data augmentation is used to increase the size and diversity of training dataset and generate total 10,080 images. Each class contained a total of 1004 images, except for the Leaf streak and tungro classes which had 1022 and 1024 images. The classes included were Bacterial leaf blight, Brown spot, Hispa, Leaf blast, Leaf scald, Leaf streak, Narrow brown spot, Sheath blight, Tungro, and Healthy. All images were in .jpg format with a resolution of 128 × 128 pixels and captured under the same illumination and white background setting. The dataset used in this work is detailed in Table 3 and some sample images of the rice leaf diseases are illustrated in Figure 2.

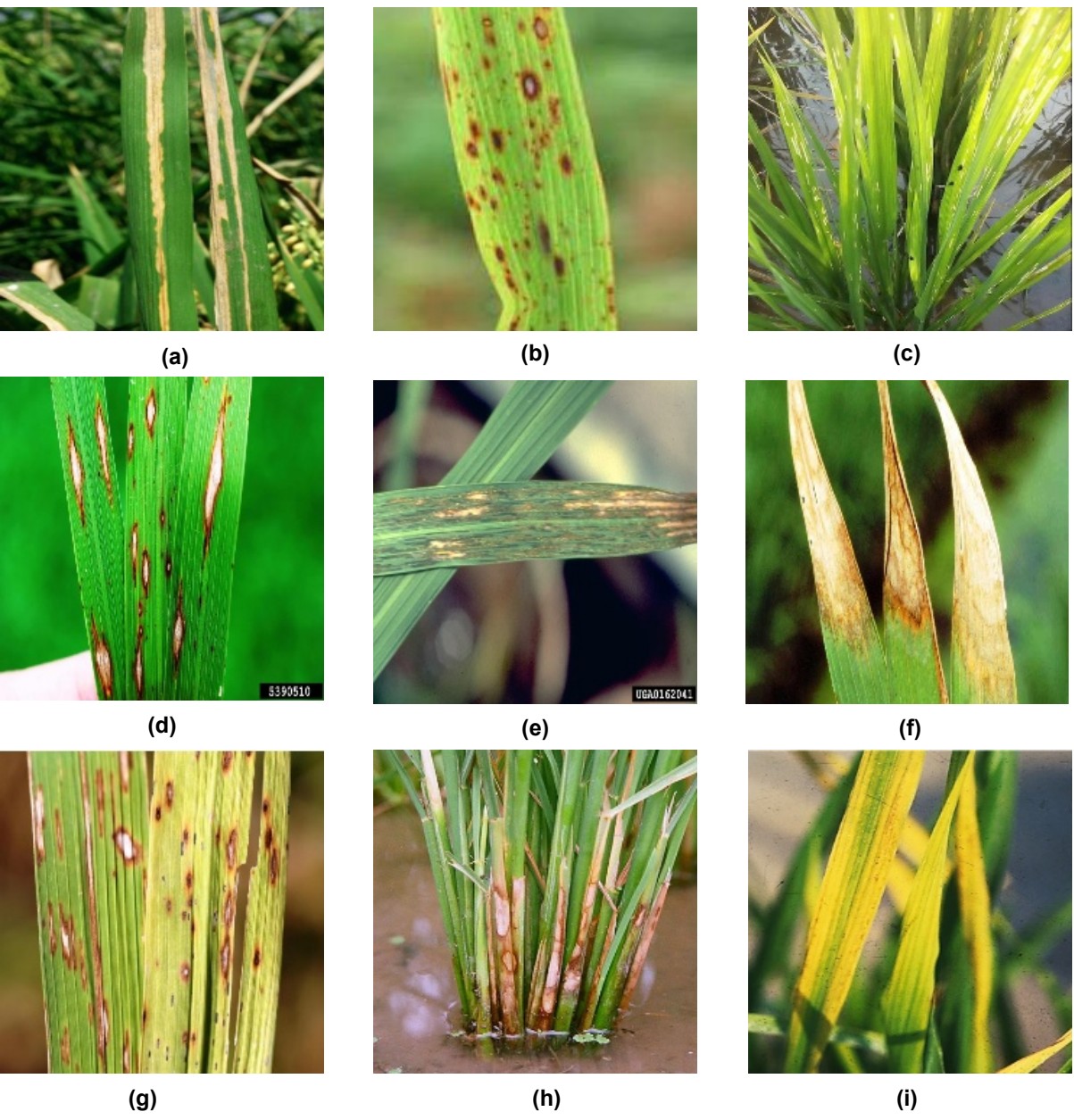

**Figure 2.** Rice leaf diseases: (**a**) Bacterial Leaf Blight (**b**) Brown spot (**c**) Hispa (**d**) Leaf Blast (**e**) Leaf Streak (**f**) Leaf Scald (**g**) Narrow Brown Spot (**h**) Sheath Blight (**i**) Tungro.

**Table 3.** Rice leaf disease dataset.

| Leaf Disease | No of Images |
|---|---|
| Bacterial_leaf_blight | 1004 |
| Brown_spot | 1004 |
| Healthy | 1004 |
| Hispa | 1006 |
| Leaf_blast | 1004 |
| Leaf_scald | 1004 |
| Leaf_streak | 1022 |
| Narrow_brown_spot | 1004 |
| Sheath_blight | 1004 |
| Tungro | 1024 |
| Total | 10,080 |

### 2.2. Preprocessing (Image Quality Enhancement and Data Augmentation)

In this work, All models were trained with identical hyperparameters as shown in Table 4. All images of leaf diseases were pre-processed by resizing them to fit the input size of the respective pre-trained CNNs, as specified in Table 5. All images were colored RGB (red, green, blue) images. To increase the amount of training data, data augmentation [33] was employed. Data augmentation helps prevent overfitting and improves model generalization. It increases the amount of training data available and enhances model robustness. This method includes image rotation, reflection, and shear parameters. Some sample augmented images were shown in Figure 3. The values of the data Augmenter are listed in Table 6. The RandXReflection was set to true to enable horizontal reflection of images. The RandXTranslation (Range of horizontal translation) and RandXTranslation (the range of vertical translation) were set to −3 to 3, allowing for a small range of translation in the images. The RandXShear (Range of horizontal shear) and RandYShear (Range of vertical shear) were set to −30 to 30, which allowed for a range of shear in the images, measured as an angle in degrees. This technique can increase the diversity of the training dataset and improve the model's performance.

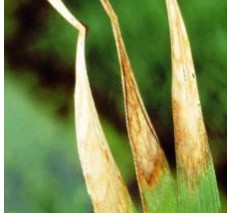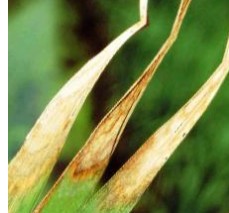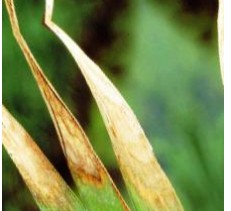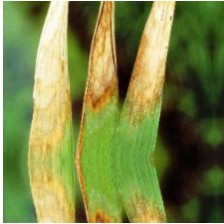

**Figure 3.** Sample augmented diseased images.

**Table 4.** Training Parameters.

| Function Name | Value |
|---|---|
| Optimizer | Adam |
| MiniBatchSize | 32 |
| MaxEpochs | 30 |
| ValidationFrequency | 30 |
| VerboseFrequency | 30 |
| ExecutionEnvironment | gpu |
| Verbose | true |
| LearnRateDropFactor | 0.1 |
| LearnRateDropPeriod | 8 |
| LearnRateSchedule | none |
| Shuffle | Every-epoch |

**Table 5.** Pre-trained models input size.

| Pre-Trained CNN Model | Parameters (Millions) | Input Image Size |
|---|---|---|
| ResNet50 | 25.6 | $224 \times 224 \times 3$ |
| ResNet101 | 44.6 | $224 \times 224 \times 3$ |
| GoogleNet | 7 | $224 \times 224 \times 3$ |
| VGG16 | 138 | $224 \times 224 \times 3$ |
| Shufflenet | 1.4 | $224 \times 224 \times 3$ |
| NasNetMobile | 5.3 | $224 \times 224 \times 3$ |
| MobileNetV2 | 3.5 | $224 \times 224 \times 3$ |
| Efficientnetb0 | 5.3 | $224 \times 224 \times 3$ |
| DenseNet201 | 20 | $224 \times 224 \times 3$ |
| AlexNet | 61 | $227 \times 227 \times 3$ |
| Squeeznet | 1.24 | $227 \times 227 \times 3$ |
| Darknet53 | 41.6 | $256 \times 256 \times 3$ |
| InceptionV3 | 23.9 | $229 \times 229 \times 3$ |
| InceptionResnetV2 | 55.9 | $229 \times 229 \times 3$ |
| Xception | 22.9 | $229 \times 229 \times 3$ |

**Table 6.** Data Augmentation.

| Properties | Values |
|---|---|
| RandXReflection | True |
| RandXTranslation | $-3$ to $3$ |
| RandYTranslation | $-3$ to $3$ |
| RandXShear | $-30$ to $30$ |
| RandYShear | $-30$ to $30$ |

### 2.3. Pre-Trained Deep Neural Network

Convolutional Neural Networks are a type of deep neural network that have become popular in recent years, particularly in agriculture. They are designed to recognize objects by using layers that include convolution, pooling, and fully connected layers. These layers work together using backpropagation to adapt and optimize the network [27]. The main objective of CNNs is to create a more extensive network with fewer parameters.

Transfer learning is a technique where we take CNN models that have already been trained on millions of images on ImageNet and continue to train them on a smaller set of images [34]. There is one way to accelerate the training process, prevent overfitting, improve model interpretability, reduce memory requirements and optimize efficiency is to use the layer freezing technique, which helps preserve the weights of initial layers and prevent them from being altered. Noor et al. [35] achieved higher accuracy from the layer freezing technique. This can be done in several ways:

Freezing all Convolutional Layer weights: A technique where only the weights of the Fully-Connected Layers are adjusted during training is achieved by freezing the weights of all Convolutional Layers, and replacing the Fully-Connected Layers of the old CNN model with new Fully-Connected Layers trained on recent data.

Freezing some Convolutional Layer weights: The pre-trained weights in the subsequent Convolutional Layers and the initialized weights in the personalized fully-connected layers are adjusted during training.

Unfreezing all Convolutional Layer weights: All of the weights in the convolutional layers are unfrozen while the fully connected layers are eliminated from the original CNN model.

Selecting the best pre-trained model for image classification depends on several factors, including the size of the dataset, computational constraints, and the specific requirements of the task. It's important to experiment with different models and fine-tuning techniques to find the best fit for our particular use case.

The goal of this study is to investigate how pre-trained deep learning CNN models can be adapted, retrained, and used to classify rice leaf diseases from images of the leaves. The

study evaluated 15 pre-trained CNN models, which differ in terms of input size, architecture, and computational efficiency. The hyperparameters used for fine-tuning the training of these models were standardized across the study to ensure a fair comparison. The pre-trained CNN models evaluated in this study were [36]: DarkNet-53, DenseNet-201, GoogLeNet, Inceptionv3, MobileNetv2, ResNet-50, ResNet-101, ShuffleNet, SqueezeNet, Xception, InceptionResNetV2, NasNetMobile, VGG16, AlexNet, and EfficientNetB0. The study aimed to determine the most effective way to use these models for rice leaf disease classification.

### 2.4. Evaluation Setup

In our experiments, we use transfer learning models that have already been trained on the 1000 classes in the Imagenet dataset. To make these models work for our specific use case, we change the first and last layers, such as fully connected, convolutional, softmax, and classification output layers, depending on how the model is built. Generally, lower layers of pre-trained models capture general features, while upper layers capture task-specific features. By modifying the upper layers, the pre-trained model can be adapted to new tasks while leveraging the knowledge learned from the original task.Then, after all models were trained using the same hyperparameters. The training was conducted for 30 epochs, which were determined based on the models' performance during training and validation. A learning rate of 0.0001 was used, and the Adam optimizer was applied for network training. The models' capability to generalize to a more extensive testing set was evaluated through several data partitioning procedures. The first method split the dataset into 60% training and 40% validation sets. The second method used 80% of the images for training, and the third method used 90% of the images for training [37].

The models were constructed and evaluated on a LENOVO DESKTOP-QT2QLGA system, which boasts 64 GB of RAM, a powerful Intel(R) Xeon(R) W-2125 CPU clocked at 4.00 GHz, 4 cores, a high-performance NVIDIA GeForce GTX 1080 Ti graphics card, and 8 logical processors. The building and testing process was conducted using the MATLAB R2021a program.

### 2.5. Performance Evaluation Metrics

In machine learning, performance metrics are used to evaluate the accuracy and effectiveness of a model. Different types of metrics are used depending on the type of problem and the algorithm being used. In this paper we used the following classification metrics accuracy, precision, recall, f1-score specificity, Matthews Correlation Coefficient (MCC), and False Positive Rate (FPR). Accuracyis the proportion of correctly classified disease samples out of the total number of disease samples. Precision is the proportion of true positive predictions out of all positive predictions. Recall (Sensitivity or true positive rate) is the proportion of true positives out of all actual positive samples. F1 Score is the harmonic mean of precision and recall. Specificity is a performance metric that measures the ability of a model to correctly identify negative samples. MCC is a metric that takes into account true and false positives and negatives and gives a value ranging from $-1$ to 1 which shows the correlation between observation and prediction. FPR It is a measure of how often a test incorrectly detects that an event of interest has occurred when it has not.

The measures used to assess the effectiveness of CNN models are depicted in Equations (1)–(6).

$$Accuracy = \frac{T_{+ve} + T_{-ve}}{T_{+ve} + T_{-ve} + F_{+ve} + F_{-ve}} \tag{1}$$

$$Precision = \frac{T_{+ve}}{T_{+ve} + F_{+ve}} \tag{2}$$

$$Recall = \frac{T_{+ve}}{T_{+ve} + F_{-ve}} \tag{3}$$

$$F1Score = 2 \times \frac{(Preciison \times Recall)}{(Preciison + Recall)} \tag{4}$$

$$Specificity = \frac{T_{-ve}}{F_{+ve} + T_{-ve}} \tag{5}$$

$$MCC = \frac{(T_{+ve} \times T_{-ve}) - (F_{+ve} \times F_{-ve})}{\sqrt{(T_{+ve} + F_{+ve})(T_{+ve} + F_{-ve})(T_{-ve} + F_{+ve})(T_{-ve} + F_{-ve})}} \tag{6}$$

### 3. Experimental Results and Discussion

To evaluate the ability of CNN models in recognizing the disease type of rice leaves, we conducted experiments. To account for differences in the random data division of images into training and validation sets, the training process was repeated ten times. The final results were calculated by taking the average. We logged the performance evaluation metrics of all the CNN models and also documented the duration of the training and validation phases.

Three data partition methodologies were employed (i.e., 60/40, 80/20, and 90/10), which may indicate the capabilities of the various models to train from additional data as well as any underfitting/overfitting abnormalities. The first data partition scheme was 60/40, the mean overall accuracy, precision, recall, F1-score, specificity, MCC and FPR of pretrained deep neural network models were tabulated in Table 7. All models performed with extremely high precision in addition to the other evaluation metrics. Most models did remarkably well, with InceptionResNetV2 achieving the highest mean F1 score of 98.21% and specificity of 99.81%. The lowest performing model was Alexnet with mean F1 score of 86.03% and a specificity value of 98.49%. The performance values were validated by Confusion matrices for the top and lowest performing models was displayed in Figure 4.

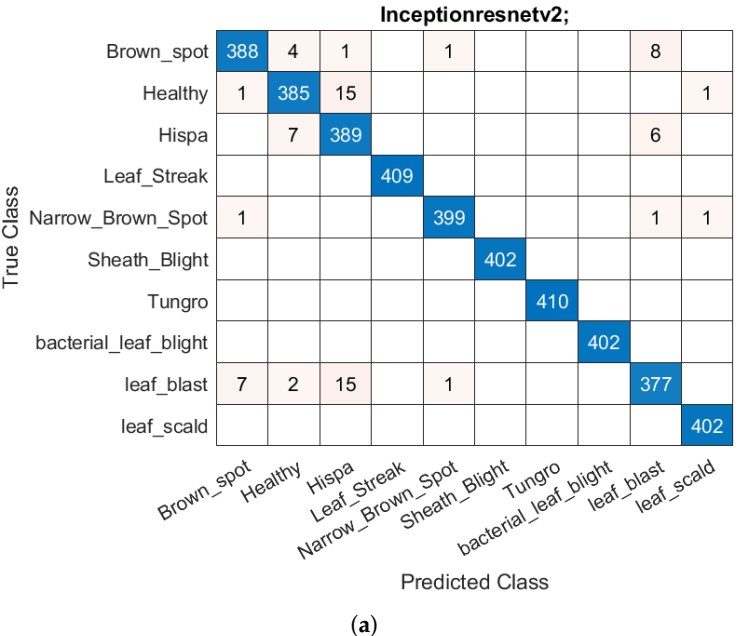

(**a**)

**Figure 4.** *Cont.*

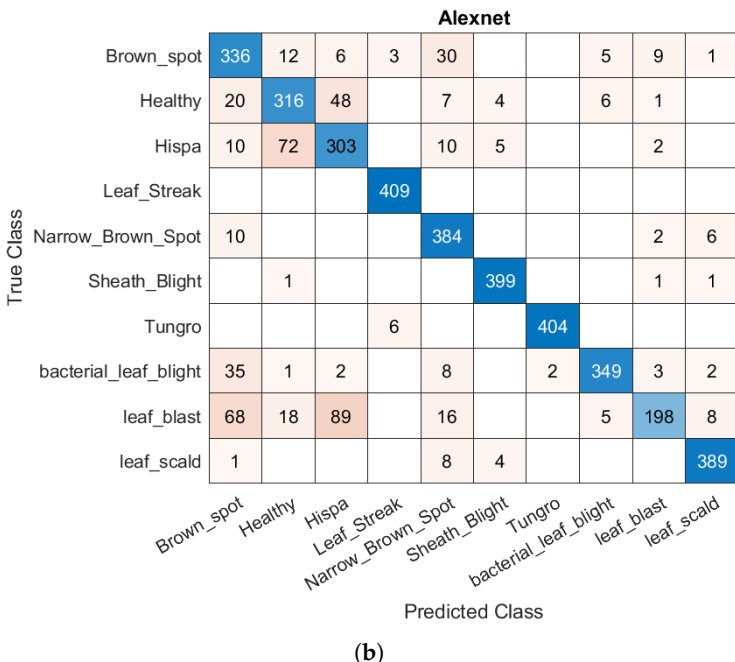

(**b**)

**Figure 4.** Confusion matrices for the finest and the poor performing pre-trained models using 60% of the training data. (**a**) InceptionResNetV2; (**b**) AlexNet.

**Table 7.** The mean overall performance metric values using using 60% of the training data.

| Model | Accuracy (%) | Precision (%) | Recall (%) | F1-Score (%) | Specificity (%) | MCC | FPR |
|---|---|---|---|---|---|---|---|
| InceptionResNetV2 | 99.64 | 98.22 | 98.2 | 98.21 | 99.81 | 0.98 | 0.017 |
| Xception | 99.61 | 98.13 | 98.08 | 98.08 | 99.78 | 0.978 | 0.018 |
| ResNet50 | 99.59 | 98.03 | 97.98 | 97.97 | 99.77 | 0.977 | 0.019 |
| InceptionV3 | 99.59 | 97.99 | 97.96 | 97.96 | 99.77 | 0.977 | 0.021 |
| DenseNet201 | 99.58 | 97.97 | 97.93 | 97.93 | 99.77 | 0.977 | 0.022 |
| EfficientNetB0 | 99.51 | 97.57 | 97.56 | 97.55 | 99.73 | 0.972 | 0.024 |
| MobileNetV2 | 99.47 | 97.53 | 97.38 | 97.38 | 99.71 | 0.971 | 0.024 |
| ResNet101 | 99.47 | 97.39 | 97.36 | 97.35 | 99.7 | 0.97 | 0.026 |
| GoogleNet | 99.35 | 96.77 | 96.76 | 96.75 | 99.64 | 0.964 | 0.032 |
| NasNetMobile | 99.27 | 96.4 | 96.34 | 96.35 | 99.59 | 0.959 | 0.035 |
| ShuffleNet | 99.19 | 96.06 | 95.94 | 95.94 | 99.55 | 0.955 | 0.039 |
| DarkNet53 | 99.18 | 96.18 | 95.89 | 95.89 | 95.54 | 0.955 | 0.038 |
| SqueezeNet | 98.98 | 95.22 | 94.92 | 94.95 | 99.43 | 0.955 | 0.047 |
| VGG16 | 98.21 | 91.18 | 90.99 | 91.01 | 99.01 | 0.901 | 0.088 |
| AlexNet | 97.28 | 87.28 | 86.37 | 86.03 | 98.49 | 0.851 | 0.127 |

According to the AlexNet confusion matrix, some images of leaf blast are misclassified as brown spot and hispa.

Table 8 presented the performance metrics for 10 runs of 15 pre-trained deep learning models, using 80% of the data for training. The performance of the models was evaluated using the remaining 20% of the data, and the results were tabulated. The efficientnetb0 model had the highest precision at 98.48% and MCC of 0.982, which outperformed the other models. In contrast, the AlexNet model had the lowest performance with a precision of 88.39% and MCC of 0.86. The highest and lowest performance of the models can also be observed in the confusion matrices, which are presented in Figure 5. The confusion matrix of AlexNet model shows that healthy and leaf blast disease samples were misclassified as brown spot and hispa, respectively.

**Table 8.** The mean overall performance metric values using 80% of the training data.

| Model | Accuracy (%) | Precision (%) | Recall (%) | F1-Score (%) | Specificity (%) | MCC | FPR |
|---|---|---|---|---|---|---|---|
| EfficientnetB0 | 99.69 | 98.48 | 98.45 | 98.46 | 99.82 | 0.982 | 0.015 |
| InceptionV3 | 99.59 | 97.99 | 97.96 | 97.94 | 99.77 | 0.977 | 0.020 |
| InceptionresnetV2 | 99.59 | 97.95 | 97.96 | 97.95 | 99.77 | 0.977 | 0.020 |
| ResNet50 | 99.58 | 97.94 | 97.91 | 97.91 | 99.76 | 0.976 | 0.020 |
| DenseNet201 | 99.56 | 97.85 | 97.81 | 97.81 | 99.75 | 0.975 | 0.021 |
| Xception | 99.54 | 97.82 | 97.71 | 97.72 | 99.74 | 0.974 | 0.021 |
| ResNet101 | 99.53 | 97.71 | 97.66 | 97.65 | 99.74 | 0.974 | 0.022 |
| Nasnetmobile | 99.51 | 97.58 | 97.56 | 97.56 | 99.73 | 0.972 | 0.024 |
| Mobilenetv2 | 99.49 | 97.57 | 97.46 | 97.46 | 99.71 | 0.972 | 0.024 |
| Darknet-53 | 99.41 | 97.16 | 97.06 | 97.02 | 99.67 | 0.967 | 0.028 |
| Shufflenet | 99.41 | 97.09 | 97.01 | 96.99 | 99.66 | 0.967 | 0.029 |
| Googlenet | 99.32 | 96.93 | 96.61 | 96.64 | 99.62 | 0.963 | 0.030 |
| Squeeznet | 99.14 | 95.93 | 95.72 | 95.69 | 99.52 | 0.953 | 0.040 |
| VGG16 | 98.71 | 93.76 | 93.53 | 93.48 | 99.28 | 0.928 | 0.062 |
| Alexnet | 97.51 | 88.39 | 87.51 | 87.01 | 98.61 | 0.861 | 0.116 |

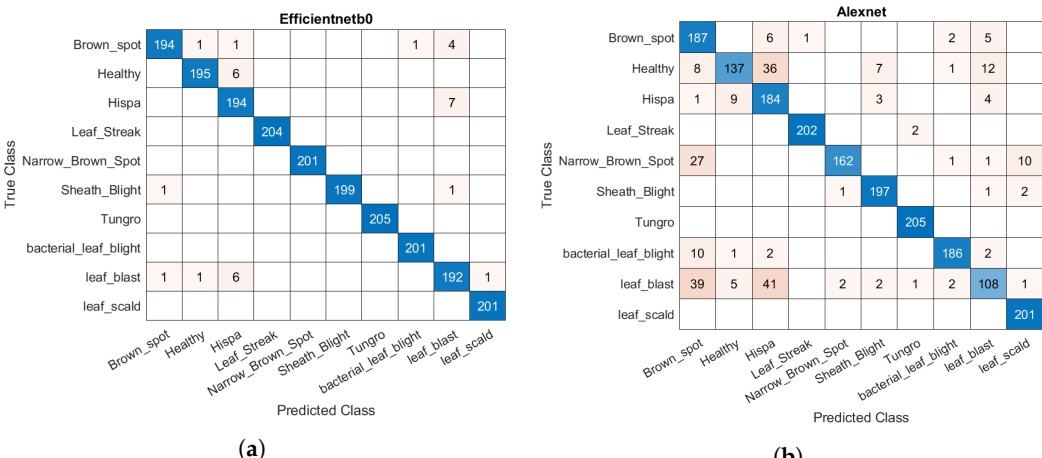

**Figure 5.** Confusion matrices for the finest and the poor performing pre-tarained models using 80% of the training data. (**a**) EfficientNetb0; (**b**) AlexNet.

The performance of 15 pre-trained models was evaluated by using random partitions of 90% of the data for training, and 10% for testing. The results were recorded and tabulated in Table 9 for each model after 10 runs. The InceptionV3 model showed exceptional performance with a precision of 98.72% and a specificity of 99.85%. On the other hand, the AlexNet model had the lowest performance, with a precision of 87.55% and a specificity of 98.48%. Figure 6 depicts the confusion matrices for the best-performing and worst-performing models. The InceptionV3 model correctly classified all diseases, however, the AlexNet model misclassified 'hispa' and 'leaf blast' as 'brown spot' and 'hispa' respectively.

Determined the mean of the performance metrics for 15 pre-trained models that were divided into three partitions (60/40, 80/20, and 90/10) as shown in Table 10.

Upon analyzing the results, it was found that the InceptionV3 model achieved the highest accuracy at 99.64%. This model also displayed impressive precision, recall, f measure, and MCC, with values of 98.23%, 98.32%, 98.20%, and 0.98, respectively. These results demonstrate that the InceptionV3 model has a high level of performance across multiple metrics. However, the AlexNet model had the lowest overall performance, with an average accuracy of 97.35%. This suggests that while the InceptionV3 model is a strong performer, the AlexNet model may not be as reliable for certain tasks.

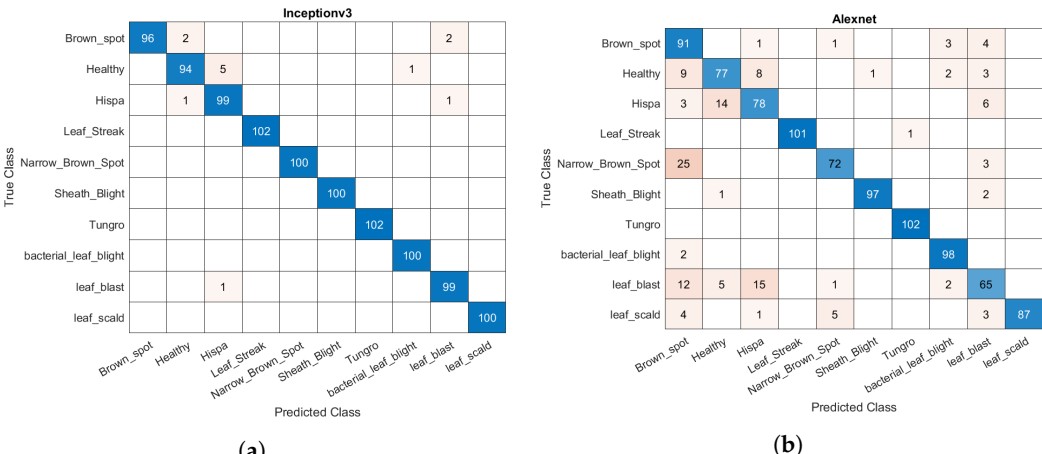

(**a**)                                                    (**b**)

**Figure 6.** Confusion matrices for the finest and the poor performing pre-trained models using 90% of the training data. (**a**) InceptionV3; (**b**) AlexNet.

**Table 9.** The mean overall performance metric values using using 90% of the training data.

| Model | Accuracy (%) | Precision (%) | Recall (%) | F1-Score (%) | Specificity (%) | MCC | FPR |
|---|---|---|---|---|---|---|---|
| InceptionV3 | 99.74 | 98.72 | 98.7 | 98.7 | 99.85 | 0.985 | 0.012 |
| DenseNet201 | 99.7 | 98.57 | 98.5 | 98.51 | 99.83 | 0.983 | 0.014 |
| Xception | 99.68 | 98.42 | 98.4 | 98.4 | 99.82 | 0.982 | 0.015 |
| Nasnetmobile | 99.66 | 98.31 | 98.3 | 98.3 | 99.81 | 0.981 | 0.016 |
| Mobilenetv2 | 99.64 | 98.25 | 98.2 | 98.2 | 99.81 | 0.981 | 0.017 |
| Shufflenet | 99.64 | 98.23 | 98.2 | 98.21 | 99.8 | 0.98 | 0.017 |
| ResNet101 | 99.64 | 98.21 | 98.2 | 98.2 | 99.8 | 0.98 | 0.017 |
| InceptionresnetV2 | 99.64 | 98.24 | 98.2 | 98.19 | 99.8 | 0.98 | 0.017 |
| Googlenet | 99.52 | 97.67 | 97.6 | 97.61 | 99.73 | 0.973 | 0.023 |
| EfficientnetB0 | 99.48 | 97.75 | 97.4 | 97.44 | 99.71 | 0.972 | 0.022 |
| Darknet-53 | 99.46 | 97.34 | 97.31 | 97.29 | 99.71 | 0.97 | 0.026 |
| ResNet50 | 99.22 | 96.34 | 96.1 | 96.09 | 99.56 | 0.957 | 0.036 |
| Squeeznet | 99.14 | 95.96 | 95.7 | 95.73 | 99.52 | 0.953 | 0.04 |
| VGG16 | 98.48 | 92.83 | 92.41 | 92.4 | 99.15 | 0.917 | 0.071 |
| Alexnet | 97.27 | 87.55 | 86.32 | 86.46 | 98.48 | 0.852 | 0.124 |

In addition, we compared the above results with K-fold cross-validation, which is a technique used to evaluate the performance of machine learning models. K-fold cross-validation is often used when the amount of available data is limited, it can still be a useful technique to consider even with sufficient data, as it helps to reduce the variance of the estimated performance and provides a more robust estimate of the model's generalization performance. The basic idea behind it is to divide the data into k subsets, called "folds," and then train the model k times, each time using a different fold as the test set and the remaining $k-1$ folds as the training set. The model's performance is then assessed by averaging the performance metrics across all k runs.

One of the key parameters in k-fold cross-validation is the value of 'K', which determines the number of folds that the data will be divided into. The larger the value of 'K', the more accurate the performance evaluation is likely to be, as more of the data is used for testing and validation. However, increasing the value of 'K' also increases the computational cost of the evaluation, as the model has to be trained and tested more times.

**Table 10.** Average performances of the Transfer learning Models.

| Model | Accuracy (%) | Precision (%) | Recall (%) | F1-Score (%) | Specificity (%) | MCC | FPR |
|-------|-------------|---------------|-----------|--------------|-----------------|-----|-----|
| InceptionV3 | 99.64 | 98.23 | 98.21 | 98.20 | 99.80 | 0.98 | 0.02 |
| InceptionresnetV2 | 99.62 | 98.14 | 98.12 | 98.12 | 99.79 | 0.98 | 0.02 |
| DenseNet201 | 99.61 | 98.13 | 98.08 | 98.08 | 99.78 | 0.98 | 0.02 |
| Xception | 99.61 | 98.12 | 98.06 | 98.07 | 99.78 | 0.98 | 0.02 |
| EfficientnetB0 | 99.56 | 97.93 | 97.80 | 97.82 | 99.75 | 0.98 | 0.02 |
| ResNet101 | 99.55 | 97.77 | 97.74 | 97.73 | 99.75 | 0.97 | 0.02 |
| MobileNetV2 | 99.53 | 97.78 | 97.68 | 97.68 | 99.74 | 0.97 | 0.02 |
| Nasnetmobile | 99.48 | 97.43 | 97.40 | 97.40 | 99.71 | 0.97 | 0.03 |
| ResNet50 | 99.46 | 97.44 | 97.33 | 97.32 | 99.70 | 0.97 | 0.03 |
| Shufflenet | 99.41 | 97.13 | 97.05 | 97.05 | 99.67 | 0.97 | 0.03 |
| GoogleNet | 99.40 | 97.12 | 96.99 | 97.00 | 99.66 | 0.97 | 0.03 |
| Darknet-53 | 99.35 | 96.89 | 96.75 | 96.73 | 98.31 | 0.96 | 0.03 |
| Squeeznet | 99.09 | 95.70 | 95.45 | 95.46 | 99.49 | 0.95 | 0.04 |
| VGG16 | 98.47 | 92.59 | 92.31 | 92.30 | 99.15 | 0.92 | 0.07 |
| AlexNet | 97.35 | 87.74 | 86.73 | 86.50 | 98.53 | 0.85 | 0.12 |

When resources are limited, it may not be possible to use a large value of 'K' without sacrificing other aspects of the model development or evaluation. In such cases, it may be necessary to set 'K' to a lower value, such as 5, to balance the trade-off between accuracy and computational cost. This means that fewer iterations of the model are used, so the results may not be as accurate as the results would have been if we have higher value of 'K' but it would be less computationally intensive. All the results should be tabulated in Table 11.

**Table 11.** 5-Fold cross validation of pre-trained models performance.

| Model | Accuracy | Specificity | Precision | Recall | F Measure | MCC | ERR |
|-------|----------|-------------|-----------|--------|-----------|-----|-----|
| InceptionResNetV2 | 98.49 | 98.56 | 88.33 | 97.81 | 92.83 | 0.921 | 0.015 |
| Xception | 98.44 | 98.65 | 88.83 | 96.52 | 92.52 | 0.917 | 0.015 |
| ResNet50 | 98.12 | 98.27 | 86.11 | 96.81 | 91.15 | 0.903 | 0.018 |
| EfficientNetB0 | 98.01 | 98.08 | 84.92 | 97.41 | 90.74 | 0.899 | 0.019 |
| ResNet101 | 97.91 | 98.06 | 84.66 | 96.62 | 90.25 | 0.893 | 0.021 |
| Inception | 97.86 | 98.13 | 85.03 | 95.42 | 89.93 | 0.889 | 0.021 |
| MobileNetV2 | 97.74 | 98.38 | 86.3 | 92.04 | 89.08 | 0.878 | 0.022 |
| NasnetMobile | 97.67 | 97.73 | 82.61 | 97.21 | 89.31 | 0.883 | 0.023 |
| Shufflenet | 97.53 | 97.67 | 82.1 | 96.22 | 88.6 | 0.875 | 0.024 |
| DenseNet201 | 97.41 | 97.53 | 81.22 | 96.32 | 88.13 | 0.871 | 0.025 |
| Darknet53 | 97.11 | 97.47 | 80.46 | 93.73 | 86.59 | 0.852 | 0.028 |
| Googlenet | 96.38 | 96.41 | 74.84 | 96.12 | 84.16 | 0.829 | 0.036 |
| Squeezenet | 96.13 | 96.22 | 73.65 | 95.32 | 83.1 | 0.818 | 0.038 |
| VGG16 | 93.36 | 92.17 | 67.87 | 78.95 | 69.42 | 0.693 | 0.089 |
| AlexNet | 87.11 | 87.83 | 42.35 | 80.61 | 55.52 | 0.523 | 0.128 |

The Figure 7 shows the confusion matrices of the best and worst-performing models. Comparing the performance results with the remaining models, InceptionResNetV2 performed at the highest accuracy of 98.49%, and 1.51% of ERR. AlexNet had performed lowest accuracy of 87.11% and 12.89% of ERR. Figure 8 displayed the 5-flod cross validation of training progress curve at the fifth fold. The Figure 8 horizontal axis represented the number of epochs to run the model and vertical axis represents accuracy and loss of the model. Figure 9 depicts the ROCs of the networks utilized for classification in this article. ROCs were used to figure out the area under the curve (AUC). Based on this data, we calculated an AUC of 99.94% for InceptionResNetV2, 99.92% for Xception, 99.91% for ResNet50, and 97.70% for Alexnet. The given information displays the ability of the CNN

to classify each class correctly using confusion matrices and ROCs, including the AUC and total accuracy of the model.

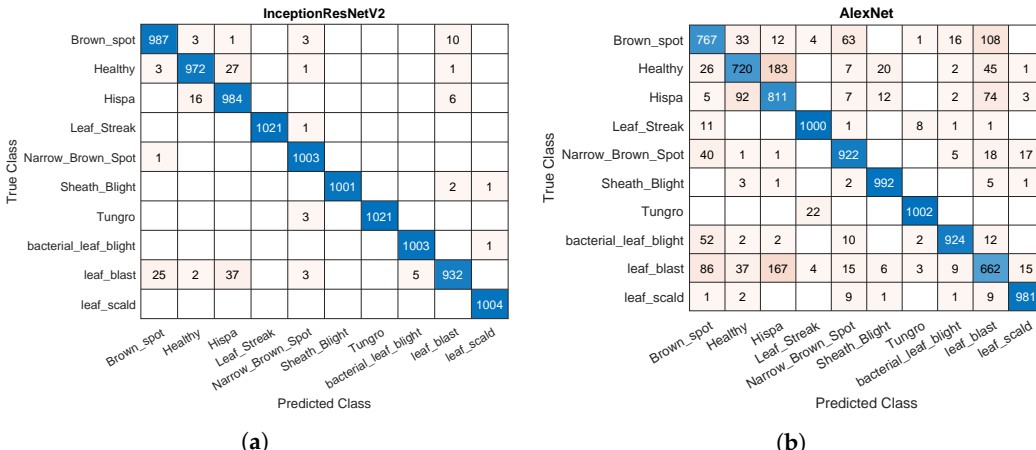

**Figure 7.** Confusion matrices for the finest and the poor performing models using 5-Fold cross validation (**a**) InceptionResNetV2; (**b**) AlexNet.

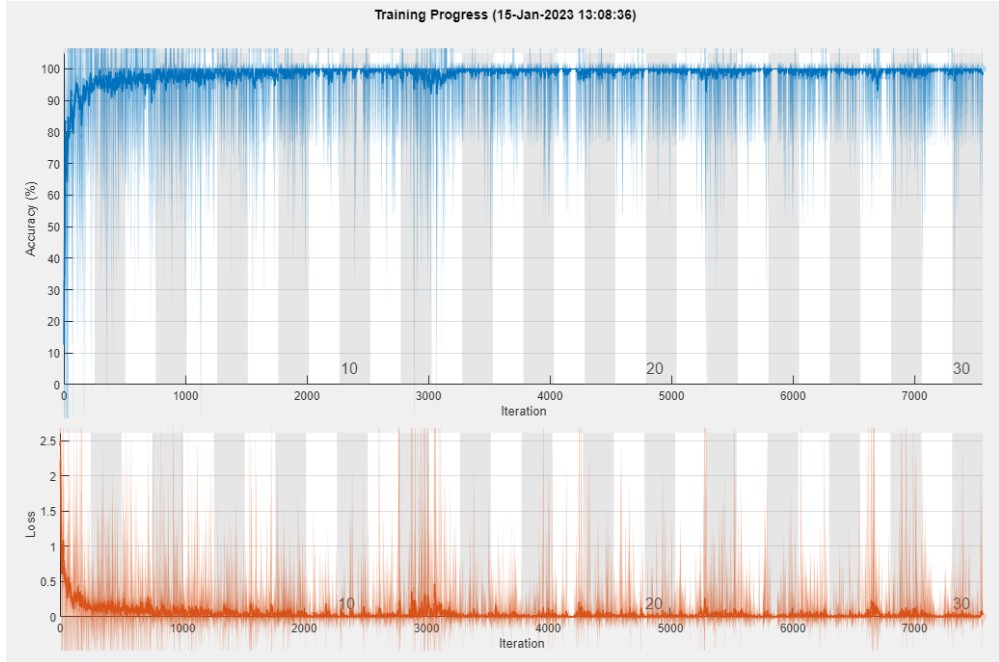

**Figure 8.** Training/Validation progress curves of InceptionResNetV2 for 5 fold cross validation.

Table 12 shows a summary of the performance of various studies on the identification and classification of rice leaf disease using machine learning and deep learning algorithms. Each study is listed with its respective accuracy percentage. The results suggest that our work achieves high levels of accuracy in identifying and classifying rice leaf diseases with an accuracy of 99.64% using the InceptionV3 model.

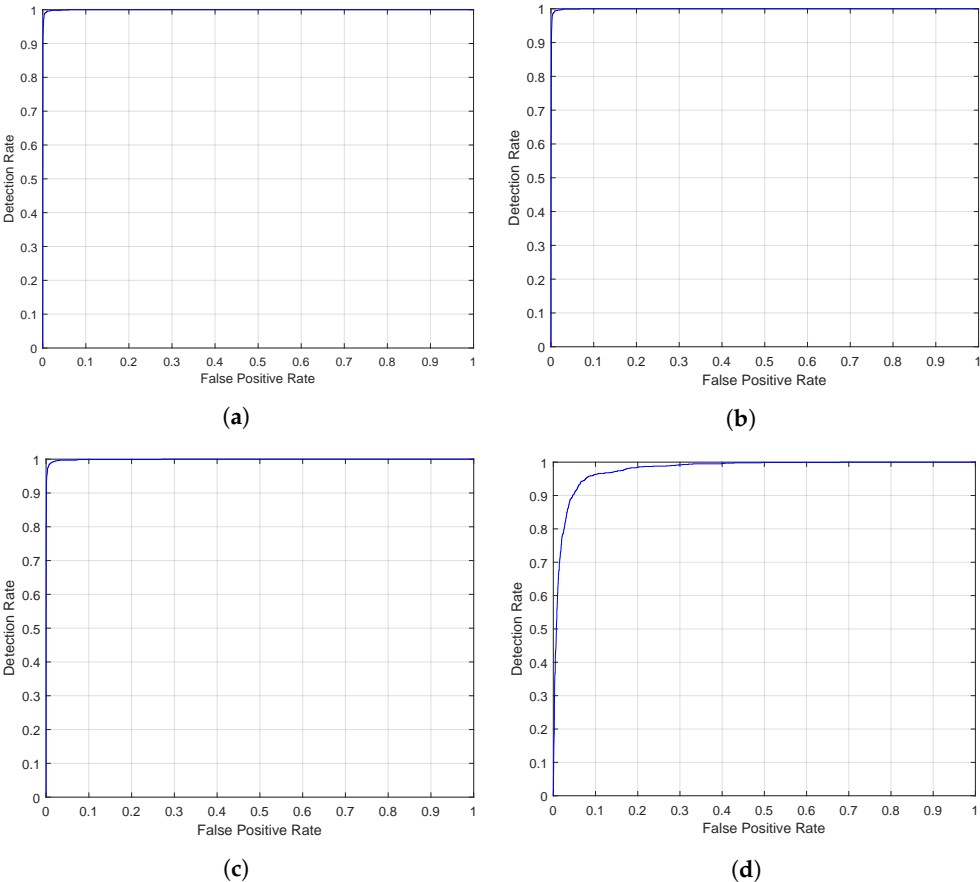

**Figure 9.** ROC Curves of the few pre-trained models using 5-Fold cross validation (**a**) InceptionResNetV2 (**b**) Xception (**c**) ResNet50 (**d**) AlexNet.

**Table 12.** Performance comparison of existing works with proposed work.

| Study | Performance (Avg Accuracy) |
| --- | --- |
| Haridasan et al. [15] | 91.45% |
| Yang et al. [16] | 99.58% |
| Latif et al. [17] | 96.08% |
| Daniya et al. [18] | 91.6% |
| Islam et al. [20] | 92.68% |
| Wang et al. [21] | 94.64% |
| Chen et al. [23] | 98.63% |
| Yakkundimath et al. [38] | 92.4% |
| Narmadha et al. [39] | 97.68% |
| Proposed Work | 99.64% |

*Limitations of This Work*

The current research on using transfer learning for rice leaf disease classification has some limitations. One of the primary challenges of using this approach is the generalizability of the models. The models used may not perform optimally on images taken from different plant species or under varying lighting conditions. This limitation can lead to incorrect classification of diseases and impact the reliability of the model. Another limitation is their ability to generalize well to different types of leaf images that are not similar to the ones it was trained on. If a model encounters images that have significant differences from the training set, it may struggle to classify the diseases accurately. If a new dataset has a skewed distribution of samples, these models may focus on the majority class and ignore the minority class, leading to poor results for the minority class. Another

limitation is that the models were only trained on a limited number of images and classes of rice diseases. As a result, it may misclassify new diseases as one of known diseases.

The models may also misclassify nutritional deficiencies as diseases. Nutritional deficiencies in rice plants can be caused by a lack of essential nutrients in the soil. The visual characteristics of nutritional deficiencies in rice plants are similar to those of diseases, but there are some key differences. For example, diseases often cause dark circles or lesions on the leaves, while nutritional deficiencies tend to cause yellowing or stunted growth. Additionally, diseases can spread from plant to plant, while nutritional deficiencies are usually caused by a lack of nutrients in the soil and affect the entire plant. It is important to differentiate between rice diseases and nutritional deficiencies, as the treatment for each will differ. Nutritional deficiencies can be addressed by adding the missing nutrients to the soil, while diseases will require specific treatments to control their spread and cure the affected plants.

While transfer learning provides a convenient solution for rice leaf disease classification, it is essential to consider the limitations of the models. Addressing these limitations through further research and experimentation can improve the performance of a model and make it more reliable for practical applications.

## 4. Conclusions and Future Directions

Rice is a vital food source for over half of the world's population and is essential for global food security. However, diseases that affect rice plants can result in significant financial losses for farmers. To address this, researchers are developing new and effective methods to combat these diseases. One promising approach is the use of advanced technologies, such as machine learning and deep learning, to identify pathogens in rice plants. This study examines the application of transfer learning to 15 pre-trained CNN models, with the goal of automating the detection of diseases in rice leaves. The results show that the InceptionV3 model was the most effective, achieving an average accuracy of 99.64%, while the AlexNet model performed poorly. It is not always possible to determine if one method is suitable for all rice varieties, as different varieties can have variations in their genetics, morphology, and physiology that may affect the manifestation of disease symptoms.

To confirm a diagnosis in such cases, it is important to consult with an expert in plant pathology and take into consideration other symptoms, such as the location and severity of the symptoms, and environmental factors, such as weather and soil conditions. Additionally, laboratory tests, such as bacterial or fungal culture, Polymerase chain reaction (PCR), and microscopy can help to confirm the identity of the pathogen.

Rice plant diseases often exhibit symptoms in different parts of the plant. For example, bacterial leaf blight is characterized by water-soaked lesions on leaves, while sheath blight is characterized by lesions on the leaf sheath and collar. Additionally, blast disease can infect the panicles and grains, leading to reduced size and quality. Therefore, by focusing only on the symptoms of the disease on the leaves, the diagnosis might not be complete. And this can lead to missing the disease at early stages. To address this, it is important to consider the symptoms of the disease on different parts of the plant in the diagnosis of the disease.

It would be beneficial to collect a more diverse dataset that includes images of different symptoms of the rice plant diseases, in order to train the model to recognize symptoms on different parts of the plant. This will enable the model to diagnose the disease more accurately and in a timely manner, which can help to prevent the disease from spreading and reduce yield loss. Additionally, to apply integrated pest management strategies for rice plants, such as regular monitoring and scouting of the field, and if in doubt, multiple samples should be taken, and the diagnostic process should be repeated. This will make the process more efficient and effective

Future works can focus on ensemble learning for rice leaf disease detection as it is considered to be more powerful than transfer learning. Ensemble learning is known for its improved performance as it combines the predictions of multiple models, reducing

variance, and bias and improving generalization, leading to better overall performance. Additionally, ensemble learning is less likely to overfit as it combines multiple models with different architectures or parameters, resulting in a more robust and generalizable model. Furthermore, ensemble learning is less affected by domain mismatch as it combines predictions from multiple models, each of which may be well-suited for different parts of the data.

Also, using Hybrid systems is an alternative solution for plant disease detection. Hybrid systems combine the strengths of multiple approaches, such as deep learning, computer vision, and expert systems, in order to improve the accuracy and robustness of the disease detection process. For example, a hybrid system may use deep learning to extract features from plant images, and then use a computer vision-based approach to classify the images based on those features. Additionally, expert systems can be incorporated to provide additional knowledge-based rules and decision-making algorithms to further improve the accuracy of the diagnosis. The combination of multiple approaches can also help to reduce the dependence on a large dataset of annotated images, making the system more versatile and adaptable to different types of plant diseases.

**Author Contributions:** C.G.S.: Methodology, Writing original draft. H.K.K.: Conceptualization, Writing, Review, Editing, Supervision. All authors have read and agreed to the published version of the manuscript.

**Funding:** This research received no external funding.

**Institutional Review Board Statement:** Not applicable.

**Informed Consent Statement:** Not applicable.

**Data Availability Statement:** Data available on request from the authors.

**Acknowledgments:** The authors would like to thank Arunkumar Gopu, Coordintor, High Performance Computing Lab at VIT-AP University, Amaravati, Andhra Pradesh, India for his support and providing resources for the Experimental analysis.

**Conflicts of Interest:** The authors declare no conflict of interest.

## Abbreviations

The following abbreviations are used in this manuscript:

| | |
|---|---|
| CNN | Convolutional Neural Networks |
| CORAL | CORrelation ALignment |
| DDC | deep domain confusion |
| DCNN | Deep Convolutional Neural Network |
| ExpRHGSO | Exponential Rider Henry Gas Solubility Optimization |
| $T_{+ve}$ | True Positive |
| $T_{-ve}$ | True Negative |
| $F_{+ve}$ | False Positive |
| $F_{-ve}$ | False Negative |
| MCC | Matthews Correlation Coefficient |
| FPR | Flase Positive Rate |

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
