# Peer review of "Automatic Recognition of Rice Leaf Diseases Using Transfer Learning"

_agronomy, doi:10.3390/agronomy13040961_

Round 1
Reviewer 1 Report
The authors used a relatively mature deep learning method to deal with the problem of rice leaf disease detection. The work is detailed and in-depth.
Here are a few suggestions:
1. How was "10080 images for experiment" collected in the paper? Are there any requirements on shooting equipment, illumination, angle, etc.? It is recommended to give an explanation in a prominent place in the text. This is important for the application of the method, and also helps readers to evaluate the robustness of the method.
2. Line 47's "that has received relatively little attention" is inaccurate.
3. Could the author comment on the adaptability of this method on different rice varieties at the conclusion?
4. Whether the data of the article can be shared on the network?
Reviewer 2 Report
1. L18: the structure of introduction is confusing and not logical. Too much deep learning and CNN introduction.
2. L199: Relevant work is listed by references and lacks necessary summary.
3.L290-292: The difference between feature extraction and implicit processing
needs to be explained.Why is implicit processing better?
4.L294-298: why is the classification layer updated first, the classification layer is at the end of the network, while the previous features have not been extracted and updated.
5.L299-302: I can't understand what operation is performed and how to add a new layer instead.
6.L338-346: the data enhancement description is vague, only the parameters and no description of the role, and no examples to be shown.
7. L358-374: these are the basic content of CNN, no need to introduce, should be specific about the methods you use, the text lacks the specific introduction of their own methods.
8. L378-387: these are several means of freezing the network, but the detailed description is missing in this section.
9.L400-402: Why do you want to change it? What is the need to be met? What are the roles of the other layers?
10.L479-480: K-fold cross-validation is usually used in the case of small amount of data, is it necessary to conduct this experiment with sufficient data?
11. References have bold words and the format seems to be inconsistent
Reviewer 3 Report
Dear Authors,
I revised the manuscript "Automatic Recognition of Rice Leaf Diseases Using Transfer Learning" submitted to Agronomy journal. The manuscript is very interesting, with well-described materials and methods. The results of the research are presented in detail. The limitations of the work are well described. Unfortunately, the weakest part of the manuscript is the discussion of results.
In addition, I have some concerns which need to be addressed before considering for final publication.
Minor comments:
1. The section "1. Introduction" and the section "2. Related Work" should be combined into one new section "1.Introduction". In addition, the section "2. Related Work" is too long and should be shortened when the two sections are merged.
2. Lines 537-573. This part of the conclusions is too long and should be shortened. There is a repetition of content from the abstract and introduction. This section should describe the main results and conclusions of this work.
Major comment:
Section "4. Experimental Results and Discussion". This section describes the results of the experiment very well. Unfortunately, a real discussion of the results needs to be included. You need to compare your results to the results from other papers. Additional references are also needed.
Reviewer 4 Report
Authors proposed a study transfer learning approach on pre-trained CNN models for the automatic identification of Rice leave diseases. The work is promising. The only observation is that the methodology used to be included in the summary. Being a work related to image data, it is requested that you include citations about other similar databases such as RoCoLe, LeLepHid, PlantVillage, PlantPathDB, etc.
